# Psychosocial Experiences of HIV-Positive Women of African Descent in the Cultural Context of Infant Feeding: A Three-Country Comparative Analyses

**DOI:** 10.3390/ijerph17197150

**Published:** 2020-09-29

**Authors:** Josephine Etowa, Hilary Nare, Doris M. Kakuru, Egbe B. Etowa

**Affiliations:** 1School of Nursing, Faculty of Health Sciences, University of Ottawa, Ottawa, ON K1H 8M5, Canada; 2Canadian of African Descent Health Association, Ottawa, ON K1H 8M5, Canada; hnare@uottawa.ca; 3School of Child and Youth Care, University of Victoria, Victoria, BC V8P 5C2, Canada; doriskakuru@uvic.ca; 4Department of Sociology, Anthropology & Criminology, Faculty of Arts, Humanities & Social Sciences, University of Windsor, Windsor, ON N9B 3P4, Canada; eetowa@uwindsor.ca

**Keywords:** black mothers, infant feeding guidelines, breast feeding, sociocultural factors, women living with HIV, mother-to-child transmission, psychosocial influences

## Abstract

Infant feeding among mothers of African descent living with Human Immunodeficiency Virus (HIV) is a critical practice that is influenced by policies, cultural expectations, and the resultant psychosocial state of the mother. Hence, this paper draws insights from a broader infant feeding study. It provides insights into how guidelines on infant feeding practices, cultural expectations, migration, or geographic status intersect to influence the psychosocial experiences of mothers living with HIV. We compared psychosocial experiences of Black mothers of African descent living with HIV in Nigeria versus those in high-income countries (Canada and USA), in the context of contrasting national infant feeding guidelines, cultural beliefs about breastfeeding, and geographic locations. Survey was conducted in venue-based convenience samples in two comparative groups: (Ottawa, Canada and Miami-FL, USA combined [*n* = 290]), and (Port Harcourt, Nigeria [*n* = 400]). Using independent samples t-statistics, we compared the means and distributions of six psychosocial attributes between Black mothers in two distinct: Infant feeding groups (IFGs), cultural, and geographical contexts at *p* < 0.05. Psychosocial attributes, such as discrimination and stigma, were greater in women who exclusively formula feed (EFF) than in women who exclusively breastfeed (EBF) at *p* < 0.01. Heightened vigilance, discrimination, and stigma scores were greater in women whose infant feeding practices were informed by cultural beliefs (CBs) compared to those not informed by CBs at *p* < 0.001. Discrimination and stigma scores were greater among mothers in Canada and the USA than in Nigeria at *p* < 0.001. Heightened vigilance and perceived stress scores were less among women in Canada and the USA than in Nigeria at *p* < 0.001. The guidelines on infant feeding practices for mothers with HIV should consider cultural expectations and migration/locational status of mothers.

## 1. Introduction

HIV-positive mothers of African descent living in different cultural and geographic contexts with contrasting national infant feeding guidelines experience varied perspectives toward breastfeeding. To understand the varying perspectives, this study examined mothers who exclusively breastfeed versus those who exclusively formula feed their infants, in line with the respective infant feeding recommendations in their countries of residence. The dichotomy of contemporary infant feeding choices and cultural expectations for seropositive Black mothers can lead to stress and confusion in the Afrocentric societies. Women of African descent living with HIV who must make decisions about infant feeding methods must navigate deeper complexities embedded in the cultural expectations–national guidelines terrain.

Given that, about half of the world’s HIV cases are women of childbearing age, the elimination of vertical HIV transmission from mother to newborn is recognized as a global health priority. The Global Plan to HIV, launched in 2011 by the Joint United Nations Program on HIV/AIDS (UNAIDS), aimed to reduce mother-to-child transmission (MTCT) and included routine prenatal screening, mother and infant antiretroviral prophylaxis, caesarean section, and avoidance of breastfeeding [1,2], since breastfeeding is a major route of MTCT [3]. In 2010, the World Health Organisation (WHO) recommended that mothers living with HIV continuously breastfeed until the infant is 12 months old, provided that mother and/or baby received antiretroviral therapy (ART) at the same time. However, this guideline still recommends avoiding breastfeeding if formula feeding was accessible, feasible, affordable, sustainable, and safe (WHO guideline update, 2016). National governments were asked to decide to align their health services with only one of these approaches. Consequently, the implementation of the guidelines varies globally. For instance, while high-income countries such as Canada and the U.S. recommend exclusive formula feeding, low and middle-income countries such as Nigeria recommend exclusive breastfeeding [4,5]. Conflicting guidelines create challenges for Black women living in high-income countries because of the significant influence of cultural norms on mothering [6,7]. Culture not only specifies the care available to childbearing mothers, but it also socializes and educates, thereby, allowing access to this care [8].

The centrality of culture as a factor influencing infant feeding has been highlighted by various African studies, such as a Tanzanian study which quoted a mother as saying, “a real mother should breastfeed her child”, noting the perception of being a “bad mother” for not attending to their child in typical ways [9,10]. A Nigerian study found that irrespective of HIV status, some women reported that they would still breastfeed their children [11,12]. Moreover, another study, found that breastfeeding was perceived as essential to baby’s health by strengthening physical and spiritual bonds between mothers and their children [13]. A Malawian study on HIV-positive mothers revealed that people in social gatherings questioned mothers who did not breastfeed when their children cried [14] and similar sentiments were expressed by African mothers in the U.K. [15]. In contrast, HIV-positive Black women living in Canada and the U.S. face conflicting expectations about infant feeding and feel pressured by societal and personal expectations, cultural norms, and legal repercussions. In these countries, the act of breastfeeding while seropositive could lead to criminal charges for the mother [16,17], despite less than 2% aggregated population risk of vertical transmission through breastfeeding [18] and disparities in vertical transmission risk among specific subgroups [16]. This dilemma may create significant psychosocial distress for mothers. Psychological distress is a common mental health problem, described as a state of emotional suffering associated with stressors and demands that are difficult to cope with in daily life [19]. This may often result in and a sense of loss of dignity as a “real” mother, contributing to depression, stress, and hopelessness among women living with HIV. There is great tension at the intersection of the “contradictions between what societies expect of mothers and what mothers themselves do” [20]. For example, a Canadian study showed that, if HIV-positive mothers adhered to the Canadian infant feeding guidelines, they encountered difficulty maintaining their ascribed roles as respected wives, mothers, and members of the society at large [21].

## 2. Materials and Methods

This study is drawn from a larger community participatory research that studies the sociocultural contexts and experiences of infant feeding among African Caribbean Black (ACB) women in Ottawa, Canada; Miami-FL, USA; and Port Harcourt, Nigeria. The methodology including description of study sites, study population, sampling technique, inclusion criteria, recruitment strategy, and approvals by ethic boards has been documented [22]. Data were collected using a cross-sectional multi-country survey with an effective sample size of 690. Data collection including the survey instruments, the six psychometric instruments analysed, and their relevance have also been described in preceding publications of the same broader study [19,22]. Rationale for the three sites’ comparison have also been documented under the site description section of an earlier publication [22].

The six psychosocial variables of interest were measured on psychometric scales. For example, motherhood was measured on the “Being a Mother Scale (BaM-13)” [23], and stress was measured with Cohen’s perceived stress scale [24], heightened vigilance and discrimination scales [25], and the functional social support scale. The means and distributions of these six psychosocial variables were compared within pairs of target and reference groups of HIV-positive women. These were to ultimately identify the psychosocial variables that significantly influence the infant feeding approaches of HIV-positive Black women. There were three categories of target and reference group comparisons. These include the HIV-positive Black mothers who exclusively formula-fed their infants versus those who exclusively breastfed their infants; the HIV-positive Black mothers who perceived cultural beliefs as a factor of influencing infant feeding choices versus those who perceive that cultural beliefs do not influence their infant feeding choices; and HIV-positive Black mothers residing in Canada and USA versus those residing in Nigeria. At 95% confidence interval, descriptive statistics were used to compare the equality of the means of the psychometric scale scores between the target and reference groups. This was achieved with independent-samples *t*-tests. Hence, we tested the null hypotheses of no significant differences between the means of the psychosocial variables of the target and the reference groups for each level of comparison. Results were interpreted with caution after testing for the possibility of cofounding factors.

This manuscript is based on our original three-country, three-cities study funded by the Canadian Institute of Health Research (CIHR), Funding Reference # (FRN): 144831. Ethical clearance for this study was obtained from all affiliated research ethics boards (REB) of the three sites. Approval was obtained from the Health Sciences and Science Research Ethics Board at the University of Ottawa (certificate #H08-16-27) approval date 12/08/2016, renewal dates, 12/08/2017 and 12/08/2018. The Carleton University Research Ethics Board-A (CUREB-A, certificate #106300), the Social and Behavioural Institutional Review Board at Florida International University (certificate #105160), and the Research Ethics Committee at the University of Port Harcourt (certificate #UPH/CEREMAD/REC/04). Informed consent was obtained from all participants, and they signed agreements to take part in the research and were duly informed that they could withdraw from the study anytime they deemed necessary. Participants were informed of potential risks associated with study participation and that they would have limited direct benefit from participation.

## 3. Results

### 3.1. Data Description 

The research participants were women of African descent who have had at least one infant while living with HIV, and residing in either Ottawa, Canada; Miami, Florida, USA; or in Port Harcourt, Nigeria (Table 1). Given the difficulty of recruitment due to fear of revealing the HIV status of HIV-positive mothers, preliminary screening, voluntary withdrawals from the survey, and inconsistency of some survey responses, resulted in a reduction of a total number of valid surveys in Ottawa and U.S. sites. Despite these challenges, the effective response rates which was determined by the actual number of participants were (Ottawa 89% (*n* = 89)), (Miami-Fl 67% (*n* = 201)), and (Port Harcourt, Nigeria 100% (*n* = 400)) and giving a total of *n* = 690 from the three sites. 

Descriptive statistics of valid responses indicate that all participants were on ART. About two-thirds (*n* = 210, 65.4%) of the mothers exclusively breastfed their infants while on ART in Port Harcourt, Nigeria. In contrast, most (*n* = 238, 82.1%) of the mothers in Ottawa, Canada and Miami, USA practiced exclusive formula feeding. A few mothers practiced mixed feeding in Port Harcourt (*n* = 48, 15%), but this was proportionally greater than those in Ottawa and Miami (*n* = 17, 5.9%). Similarly, a few mothers chose not to report their infant feeding practices in Port Harcourt (*n* = 55, 17.1%), but this was proportionally greater than those in Ottawa and Miami (*n* = 22, 7.6%).

### 3.2. Statistical Comparisons of the Psychosocial Outcomes of Alternative Infant Feeding Choices

Table 2 provides the results of two independent-samples *t*-tests comparing the psychosocial outcomes of exclusive formula feeding versus those of exclusive breastfeeding against the null hypotheses of no significant differences in the psychosocial outcomes.

At t (590.9) = 1.6, *p* = 0.12, and CI = 95%, the difference between functional social support scores for mothers practicing exclusive formula feeding versus those for mothers practicing exclusive breastfeeding were not statistically significant. Mothers who practiced exclusive formula feeding had a mean score of 21.9 with a standard deviation of 7.7 on the functional social support scale. This score compares with that of mothers who practiced exclusive breastfeeding, a mean score of 21.0 with a standard deviation of 5.9. Thus, the null hypothesis of no significant difference in the mean scores is accepted. The result affirmed that mothers exclusively formula feeding and those exclusively breastfeeding received approximately same level of functional social support. Judging from the standard deviations and the Levene’s test, the variance of functional social support scores among the HIV-positive mothers who exclusively formula feed was significantly higher than those of mothers who exclusively breastfeed at F = 18.3 and CI = 95%. 

Contrastingly, the difference in subgroup variances of motherhood among women who exclusively formula feed their infants versus those who exclusively breastfeed was statistically significant. Thus, the SPSS provides adjustment for these unequal variances to forestall the assumption of the homogeneity of variance in the independent *t*-test on the equality of means. At t (520.2) = 11.0, *p* < 0.001, and CI = 95%, the motherhood score (M = 37.8, SD = 9.6) in women practicing exclusive formula feeding was higher than that in mothers practicing exclusive breastfeeding (M = 29.4, SD = 7.7) with a difference of 8.4. In contrast, after adjusting for unequal variance between subgroups, it was found that the difference (0.9) in heightened vigilance score of mothers who exclusively formula feed (M = 9.0, SD = 9.6) and that of those who exclusively breastfeed (M = 9.9, SD = 4.9) was not statistically significant (t [513.5] = −1.8, *p* = 0.07, and 95% CI). 

However, after the necessary adjustment for unequal variances, the discrimination score (M = 14.5, SD = 13.2) in the mothers practicing exclusive formula feeding was higher than that in mothers practicing exclusive breastfeeding (M = 7.3, SD = 9.9) with a difference of 7.7 (at t [520.4] = 7.1, *p* < 0.001, and 95% CI). Similarly, the stigma score (M = 4.6, SD = 2.2) in the mothers exclusively formula feeding was significantly higher than that in mothers exclusively breastfeeding (M = 3.8, SD = 1.8) with a difference of 0.8 (at t [519.3] = 4.3, *p* < 0.001, and 95% CI). Conversely, the perceived stress score (M = 19.2, SD = 7.2) in the mothers practicing exclusive formula feeding was lower than in mothers practicing exclusive breastfeeding (M = 21.8, SD = 4.6) with a difference of −2.6 at (t [502.9] = −5.0, *p* < 0.001, and 95% CI).

Table 3 presents comparative statistics on the psychosocial factors of mothers who perceived cultural influence on their infant feeding choices versus that of those who did not perceive such an influence. After satisfying the assumption of the equality of variances, the functional social support score (M = 23.2, SD = 6.9) in mothers perceiving cultural influence on their infant feeding method was significantly higher than that in mothers with no cultural beliefs about infant feeding (M = 21.2, SD = 6.9) with a difference of 2.0 (at t [617.0] = −3.1, *p* = 0.001, and 95% CI). Similarly, the motherhood score (M = 36.9, SD = 9.6) in mothers perceiving cultural influence on their infant feeding method was significantly higher than that in mothers perceiving no cultural influence on their infant feeding methods (M = 31.2, SD = 8.5) with a difference of 5.7 (at t [617.0] = −6.9, *p* < 0.001, and 95% CI).

Again, the hyper-vigilance score (M = 11.3, SD = 5.0) in mothers perceiving cultural influence on their infant feeding method was significantly higher than that in mothers perceiving no cultural influence on their infant feeding methods (M = 9.6, SD = 5.3) with a difference of 1.7 (at t [617.0] = −3.5, *p* < 0.001, and 95% CI). After the necessary adjustment for unequal variances, the discrimination score (M = 19.1, SD = 14.7) in mothers perceiving cultural influence on their infant feeding method was significantly higher than that in mothers perceiving no cultural influence on their infant feeding methods (M = 7.6, SD = 9.7) with a difference of 11.5 (at t [186.8] = −8.9, *p* = 0.001, and 95% CI).

Similarly, the stigma score (M = 5.3, SD = 2.3) in mothers perceiving cultural influence on their infant feeding method was significantly higher than that in mothers perceiving no cultural influence on their infant feeding methods (M = 3.8, SD = 1.8) with a difference of 1.5 (at t [204.2] = −8.9, *p* < 0.001, and 95% CI). Conversely, with similar adjustment for unequal variances, the perceived stress score (M = 18.7, SD = 7.8) in mothers having a cultural influence on their infant feeding method was significantly lower than that in mothers perceiving no cultural influence on their infant feeding methods (M = 21.7, SD = 7.8) with a difference of 3.0 (at t [186.7] = 4.4, *p* = 0.001, and 95% CI). All the results were affirmed by the Mann–Whitney U statistics, which all showed significant difference (at *p* < 0.05 or 95% CI) as in Table 3.

Table 4 consists of the results of the independent-samples *t*-test comparing the means of psychosocial factors (enablers and stressors) of the HIV-positive Black mothers residing in their country of origin (Nigeria) versus those in the diaspora (Canada and USA). Results on the psychosocial enablers show that, after the necessary adjustment for unequal variances, (i) the functional social support score (M = 20.5, SD = 6.4) in the mothers residing in Nigeria was lower than that in mothers residing in Canada or USA (M = 22.7, SD = 8.1) with a difference of 2.2 (at t [529.1] = −3.8, *p* < 0.001, and 95% CI); (ii) the motherhood score (M = 28.6, SD = 7.1) in the mothers residing in Nigeria was significantly lower than that in mothers residing in Canada or USA (M = 38.7, SD = 9.8) with a difference of 2.2 (at t [496.1] = −14.9, *p* = 0.001, and 95% CI).

The results on psychosocial stressors show that (i) with the assumption of homogeneity of variance satisfied, the hyper-vigilance score (M = 10.2, SD = 5.2) in HIV-positive Black mothers living in Nigeria was higher than that in mothers living in Canada or USA (M = 9.2, SD = 5.5) with a difference of 1.0 (at t [687.0] = 2.4, *p* = 0.02, and 95% CI); (ii) conversely and after adjusting for unequal variances, the discrimination score (5.8 + 7.6) in HIV-positive Black mothers living in Nigeria was lower than that in mothers living in Canada or USA (17.1 + 14.0) with a difference of 11.3 (at t [411.6] = −12.4, *p* < 0.001, and 95% CI); (iii) after adjusting for unequal variances, the perceived stress score (M = 21.7, SD = 4.9) in HIV-positive Black mothers living in Nigeria was higher than that in mothers living in Canada or USA (M = 18.8, SD = 7.6) with a difference of 2.9 (at t [687.0] = 2.4, *p* = 0.02, and 95% CI). All the results were affirmed by the Mann–Whitney U statistics, which showed significant difference (at *p* < 0.05 or 95% CI, in Table 4) in the distribution of the scores on psychosocial variables across the two subgroups (mothers living in Nigeria versus mothers living in Canada or USA) that were compared.

The preceding results need to be interpreted with caution as the differences that were found between the subgroups can as well be associated with other cofounding factors. For example, results of independent-samples *t*-tests comparing variables between mothers in the North American and African cities show statistical significant difference in age (t [647.0] = 2.27, *p* = 0.02), years of formal education (t [674.0] = −11.96, *p* < 0.001), and period since HIV diagnosis (t [585.0] = 12.95, *p* < 0.001). Hence some level of differences in subgroups statistics may as well be attributed to these variables.

## 4. Discussion

We discuss the results of our analyses in comparison to existing literature under three sub-themes. These include breastfeeding as a symbol of being a good mother, cultural expectations regarding infant feeding guidelines, and the influence of hyper-vigilance and associated stress on guideline adherence.

### 4.1. Is Breastfeeding a Symbol of Being a “Good Mother” among Black Women Living with HIV?

We found no difference in motherhood scores between HIV+ Black women who exclusively breastfed and those who exclusively formula fed their infants. Hence, although cultural beliefs foster the idea that motherhood is strengthened by breastfeeding, it is likely not the case for Black women living with HIV. This may be due to several reasons with the fear of vertical transmission of HIV to infants via breastmilk being a key factor. The mother–infant bonding (or motherhood experience) cannot occur meaningfully in an environment of fear and uncertainties. Given this, many studies found that mothers expressed fear over potentially exposing their babies to HIV through breastmilk [26,27]. The fear of poisoning their babies from the misconception about breastmilk being unsafe is another serious obstacle to strengthening motherhood [14,28,29,30]. Studies have shown that some healthcare workers were sources of this misinformation that breastmilk from HIV-positive mothers is “toxic”, “bad”, or “poisonous” [27,31,32,33].

First-time mothers in the United States attach symbolic meanings to infant food and feeding-related consumer items [34]. Their study identified broad categories of baby-oriented consumerism that are the qualities and characteristics mothers sought for their babies through feeding-related consumer behaviors. They also found mother-oriented consumerism, that is, qualities and characteristics mothers sought for themselves through consumer behaviors. Baby-oriented consumerism included health, comfort, taste, and development, and mother-oriented consumerism included knowledge/control, compliance, convenience, frugality, relationships, and self-image. Thus, if satisfying baby-oriented consumerism and mother-oriented consumerism fulfills the enumerated qualities/characteristics of the baby and those of the mother simultaneously, then the goal that is traditionally accomplished through breastfeeding can be reasonably met. This view is further corroborated by the finding that motherhood is more than just breastfeeding or formula feeding but more about the various needs being met, aside from the inherent need to feed the baby. For instance, if breastfeeding is inconvenient for an HIV-positive mother for example, because of extreme fears of MTCT, then going ahead to breastfeed thwarts motherhood, as it does not satisfy her need to dispel fears. A mother might choose to breastfeed because she feels it is best for her child’s health but also because it is a free source of nourishment that facilitates bonding with baby [35]. Conversely, a mother who chooses to formula-feed might do so because her baby gains more weight while on it, as well as because of its perceived convenience. A study of U.K. mothers concluded that, far from being an “empowering” act, breastfeeding may have become more of a “normalized” moral imperative that many women experience as anything but liberational [36]. Another study of African-American mothers in the Southern United States found that 50 percent of mothers who formula feed perceived themselves to be good mothers while the other half expressed doubt about formula feeding as a symbol of motherhood [37]. Both groups acknowledged structural barriers and personal circumstances that prevent some mothers from breastfeeding [37]. Thus, the study concluded that, although a powerful cultural association between breastfeeding and good mothering is evident, it is not uniform across race and class. Similarly, as in this study, although breastfeeding is promoted as a symbol of good motherhood, it is not yet clearly the case for Black mothers living with HIV either in the diaspora or those in their origin in Africa.

### 4.2. Cultural Expectations Regarding Infant Feeding Practices

The cultural expectation among people of African descent is that a mother will breastfeed her infant. This expectation does not necessarily imply exclusive breastfeeding, as in many African cultures mixed feeding (breastmilk and replacement feed) is encouraged.

Several studies demonstrate the centrality of culture in dictating the infant feeding practices of women of African descent. A study of African immigrant mothers in Canada found that the desire to breastfeed was a common existential perception shared by African women, including those living with HIV [38,39,40]. From the cultural perspective, a Tanzanian study quoted a mother as saying, “a real mother should breastfeed her child” implying that not feeding the baby in the culturally accepted way makes a bad mother [9]. Another study established the perception that breastfeeding is essential to the baby’s health by strengthening physical and spiritual bonds between mothers and their children [11]. Extensive literature review has shown cultural values or beliefs as an explanatory factor of infant feeding choices within the African people [41].

The centrality of culture in deciding infant feeding options is a critical factor for HIV-positive Black mothers to adhere to their national policies. Non-adherence to guidelines on infant feeding by mothers living with HIV was found to be influenced by sociocultural challenges among other factors [42]. A study in Zimbabwe found that 58.8% of HIV-positive mothers considered sociocultural acceptability in choosing an infant feeding option [43]. In Kakamega county of Kenya, cultural beliefs and taboos had a strong influence on infant feeding and undermine optimal infant feeding practices [44].

Although the African culture advocates breastfeeding, it does not seem to promote exclusive breastfeeding, which is the approved recommendation for mothers living with HIV for at least the first six months of life of their infants. A study in Nigeria found that pressure associated with cultural practices led to the administration of water and herbal medications [45]. Studies in Murogoro, Tanzania and in rural south Western Nigeria found that some traditional beliefs, practices, and rites encourage the use of prelacteal feeds, as well as giving extra water, herbs, and “teas” to breastfeeding babies [45,46]. Moreover, feeding the infant water is also regarded as a cultural gesture to welcome the child into the world [47]. A mixed feeding option, however, increases the risks of HIV transmission from mother to infant among mothers living with the virus.

### 4.3. Heightened Vigilance and Associated Stress Will Force Women to Breastfeed to Dispel Any Suspicion of Being HIV Positive

Heightened vigilance was moderate among all the mothers but was higher among mothers in Nigeria compared to that in those in the North American countries. Heightened vigilance among mothers living with HIV presents as a state of being highly or abnormally alert to potential external threats related to public reactions to their HIV status and/or infant feeding practices. The feeling of being judged or unwelcomed is a common trigger of heightened vigilance among women living with HIV, especially if they perceive that their social network know their HIV status, or if they feel that others are suspicious of their HIV-positive status. In some instances, heightened vigilance may be advantageous in terms motherhood attributes if it triggers alertness to baby’s cues. However, when heightened vigilance becomes too intense and obvious, it causes anxiety, especially when it leads to impulsive behaviours [22]. Hence, heightened vigilance can pose significant constraint to the infant feeding decision-making ability of mothers living with HIV.

Women often expressed difficulties deciding not to breastfeed because it often led to alienation and stigmatization. They also find themselves in another world of being, where they likely experience the psychosocial effects of not being able to breastfeed their babies to avert MTCT of HIV, or of doing so with a burden of guilt about potentially infecting their infants with HIV. This tension might challenge their core identity as women, and as mothers, thereby limiting their sense of self-worth and identity as simply a person living with HIV, not fully experiencing the joys of motherhood, yet burdened with the challenges, of the other dimensions in which their lives intersect.

Moreover, a Malawi study of HIV-positive mothers found that people in social gatherings questioned mothers who did not breastfeed when their children cried [14]. In contrast, women (in Canada and the United States) are at risk for prosecution when potentially transmitting HIV to infants [14]. Despite less than 2% aggregated population risk of vertical transmission [18], the potential for prosecution remains in both countries, and there are disparities in vertical transmission risk among specific subgroups [16]. This contradiction creates significant stress for mothers who feel the pressure from both societal and personal expectations. There is great tension at the intersection of the “contradictions between what societies expect of mothers and what mothers themselves do” [20]. HIV-positive Black women living in Canada and the USA may be caught between conflicting cultural expectations about infant feeding. This is the case for mothers living with HIV, who would prefer to breastfeed their babies despite living in countries such as Canada and the USA, where they could be criminally charged. Women in the local ACB community in Ottawa have acknowledged this tension and subsequently approached the lead investigator requesting research on how to address this tension. Conflicting guidelines creates a lot of tension for Black women especially those living in Western countries because of the significant influence of cultural norms on mothering [6,7].

#### 4.3.1. Strengths and Limitations

Clearly, the findings reflect the differing psychosocial experiences of mothers living with HIV with respect to infant feeding in one African and two North American cities. It also, indicates certain psychosocial experiences that occurred at the same rate in comparative locations. A few mothers withheld or chose not to provide information. However, we checked for the pattern of missing values in our data to be sure that vital information is not overlooked in our interpretation. Again, because the datasets used for this analysis are cross sectional, caution is taken not to infer a cause-and-effect relationship in the analysis of the dataset. Furthermore, because we did bivariate comparison of variables, cognizance we took cognizance of relationships that may be uncounted for in the analyses.

#### 4.3.2. Implications for Interventions and Further Research

Our results highlight the significance of paying close attention to psychological factors in program design. Policy, program, and guideline development and implementation must consider strategies that address psychological distress especially among vulnerable communities such as the Black communities. For example, while providing infant formula to women living with HIV, is a great start to addressing their infant feeding needs, the availability of formula does not itself address maternal stress associated with peer pressure and cultural expectations. These expectations dictate that “good” Black mothers are expected to breastfeed unless they have “HIV”. In addition, studies exploring the effects of psychosocial factors on motherhood and infant feeding choices are limited. Moreover, psychosocial factors such as heightened vigilance have received little attention in terms of research even though they are potentially significant factors in the mental health of women living with HIV. Where such cross-sectional analyses exist, they are localized and cross sectional. There need to be studies that compares psychosocial factors related to the migrant women’s origin culture and the culture of their new locations. Such comparative analyses, comparing over time and space, are required for more effective policy-making, hence targeted and increased research funding opportunities to address this need are imperative.

## 5. Conclusions

Guidelines on infant feeding practices, cultural expectations, migration, or geographic status have intersecting effects on the psychosocial experiences of mothers living with HIV. Hence, WHO and national infant feeding guidelines or recommendations for mothers with HIV should give due considerations to cultural contexts, expectations, and geographical factors that influence psychosocial experiences of mothers living with HIV, who are confronted with difficult decisions on infant feeding. Social support, strengthening motherhood experience, spirituality, self-advocacy, and personal resilience were identified coping strategies for Black mothers facing HIV-related psychosocial distress.

## Figures and Tables

**Table 1 ijerph-17-07150-t001:** Sociodemographic characteristics of the study participants.

Sociodemographic Characteristics	Nigeria (%)	Canada and US *n* (%)
Age in years (m ± SD)	34.3 ± 5.9	33.6 ± 6.3
Relationship status:		
Single, separated, divorced, or widowed	185 (27.0)	129 (45.1)
Married	490 (71.5)	150 (52.5)
Household size (m ± SD)	4.3 ± 3.7	3.7 ± 1.7
Number of children since being HIV+ (Min, Max)	1, 5	1, 3
Years since being diagnosed HIV+ (m ± SD)	8.2 ± 5.6	11.9 ± 6.9
Education:		
Years of formal education (m ± SD)	13.2 ± 2.5	14.5 ± 1.5
Attended primary school	41 (10.4)	1 (0.4)
Attended high school or more	353 (89.6)	282 (99.6)
Main source of income:		
Wages or salaries	68 (20.6)	115 (39.7)
Self employment, informal trades, investments, etc.	242 (79.4)	28 (10.5)
Social assistance, employment insurance, pensions, etc.	0 (0)	135 (50.8)
Infant feeding practices		
* Exclusive breastfeeding	210 (65.4)	15 (5.2)
Mixed feeding	48 (15.0)	17 (5.9)
Exclusive formula feeding	57 (17.8)	238 (82.1)
Chose not to answer	55 (17.1)	22 (7.6)

*n* (%) = frequency (percent) of valid responses, m = mean, SD = standard deviation. * Exclusive breastfeeding is for minimum of the first six months of the baby’s life while on antiretroviral treatment.

**Table 2 ijerph-17-07150-t002:** Comparison of psychosocial factors in mothers practicing exclusive formula feeding and those practicing exclusive breastfeeding.

Psychosocial Factors (Measured on Psychometric Scales)	Categories of Mothers	*n* (m ± SD)	*t*-Stat. Equal Means	Deg. of Freedom	*p* (2-Tailed)
Functional Social Support	EFF	295 (21.9 ± 7.7)	1.57	521.0	0.12
	EBF	228 (21.0 ± 5.9)			
Motherhood	EFF	295 (37.8 ± 9.6)	11.03	520.2	<0.001
	EBF	228 (29.4 ±7.7)			
Heightened Vigilance	EFF	295 (9.0 ± 5.7)	−1.82	513.4	0.07
	EBF	228 (9.9 ± 4.9)			
Discrimination	EFF	295 (14.5 ± 13.2)	7.07	520.4	<0.001
	EBF	228 (7.3 ± 9.9)			
Stigma	EFF	295 (4.6 ± 2.2)	4.35	519.3	<0.001
	EBF	228 (3.8 ± 1.8)			
Perceived Stress	EFF	295 (19.2 ± 7.2)	−5.02	502.9	<0.001
	EBF	228 (21.8 ± 4.6)			

EFF = mothers who exclusively formula fed their infants, EBF = mothers who exclusively breastfed their infants while on antiretroviral treatment.

**Table 3 ijerph-17-07150-t003:** Comparison of psychosocial experiences of mothers influenced by cultural beliefs in their infant feeding practices versus those not influenced.

Psychosocial Factors (Measured on Psychometric Scales)	Categories of Mothers	*n* (m ± SD)	*t*-Stat. Equal Means	Deg. of Freedom	*p* (2-Tailed)
Functional Social Support	Not influenced	472 (21.2 ± 6.9)	−3.148	617	0.002
	Influenced	147 (23.2 ± 6.9)			
Motherhood	Not influenced	472 (31.2 ± 8.5)	−6.869	617	<0.001
	Influenced	147 (36.9 ± 9.6)			
Heightened Vigilance	Not influenced	472 (9.6 ± 5.3)	−3.526	617	0.002
	Influenced	147 (11.3 ± 5.0)			
Discrimination	Not influenced	472 (7.6 ± 9.7)	−8.881	186.842	<0.001
	Influenced	147 (19.1 ± 14.7)			
Stigma	Not influenced	472 (3.8 ± 1.8)	−7.034	204.217	<0.001
	Influenced	147 (5.3 ± 2.3)			
Perceived Stress	Not influenced	472 (21.7 ± 5.1)	4.429	186.666	<0.001
	Influenced	147 (18.7 ± 7.8)			

**Table 4 ijerph-17-07150-t004:** Comparison of psychosocial experiences of mothers residing in Canada or U.S. with those living in Nigeria.

Psychosocial Factors (Measured on Psychometric Scales)	Mother’s Country of Residence	*n* (m ± SD)	*t*-Stat. Equal Means	Deg.of Freedom	*p* (2-Tailed)
Functional Social Support	Nigeria	400 (20.5 ± 6.4)	−3.812	529.1	<0.001
	Canada or USA	289 (22.7 ± 8.1)			
Motherhood	Nigeria	400 (28.6 ± 7.1)	−14.947	496.074	<0.001
	Canada or USA	289 (38.7 ± 9.8)			
Heightened Vigilance	Nigeria	400 (10.2 ± 5.2)	2.37	687	0.02
	Canada or USA	289 (9.2 ± 5.5)			
Discrimination	Nigeria	400 (5.8 ± 7.6)	−12.411	411.604	<0.001
	Canada or USA	290 (17.1 ± 14.0)			
Stigma	Nigeria	400 (3.8 ± 1.6)	−7.257	477.322	<0.001
	Canada or USA	290 (5.0 ± 2.4)			
Perceived Stress	Nigeria	400 (21.8 ± 4.9)	5.886	457.446	<0.001
	Canada or USA	290 (18.8 ± 7.6)			

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
