# Peer review of "Psychosocial Experiences of HIV-Positive Women of African Descent in the Cultural Context of Infant Feeding: A Three-Country Comparative Analyses"

_ijerph, 2020, doi:10.3390/ijerph17197150_

Round 1

Reviewer 1 Report

This is a relevant work about infant feeding practices among HIV positive black women living in Nigeria, USA and Canada.

Readers would benefit from a concise explanation about "Heightened vigilange". 

The acromym ACB that appears at first time at line 89 should be fully described.

Author Response

All comments have been addressed in the document attached below. 

Reviewer 2 Report

  • Throughout article if using any abbreviations for the first time, authors should elaborate them and put abbreviations in brackets. E.g. MTCT.
  • Table 1-4: Authors should add parameters where it is applicable e.g. age in years.
  • Table 2-4: ‘t’ stat and degree of freedom are not required to be shown. Authors should keep either an F stat or t stat and related p value in all tables whichever they think is important. If authors would like to give more information, they can provide those tables as supplementary info or also just provide them in description of the table.
  • Strengths and limitations: Authors must provide this section at the end of the discussion. Authors have provided some limitations at the end of the results. But research is based on a survey and it was cross sectional. Survey method, and using made up scores have inherent tendency for measurement error and cross-sectional studies are not reliable for cause and effect association. Authors should comment about this in limitations section. Results in the study might have important implications, but method of the study makes them prone to have some significant limitations. Accepting this in limitations is important.

Author Response

All reviewer comments have been addressed. Please find our responses attached in the document below.  

Reviewer 3 Report

Thank you for this interesting paper that I enjoyed reading. It provides some important findings on an important topic.

It’s written well and clearly structured, though I’d probably recommend some final copy editing to cut down on any overwriting, especially in the discussion section. I would also consider shortening the abstract a little.

I have some comments and recommendations which I hope are helpful.

INTRODUCTION: This provides useful background information about issues around HIV positive mothers of African descent and infant feeding. It identifies some of the difficulties and impact of guidelines on the lived experience of being a mother in this group. Citations are current.

I would recommend removing the two sentences starting ‘This study…’ (line 38) and finishing ‘countries of residence’ (line 41). You don’t need this here.

A stronger definition of ‘psychosocial distress’ would be helpful in the introduction.

What is ‘ACB’ (line 89)?

Also, I wasn’t entirely sure about this sentence (lines 41-43): “The dichotomy of behaviour and culture with regards to infant feeding by seropositive Black mothers can lead to misrepresentation of facts in Afrocentric societies.” What do you actually mean here?

MATERIALS AND METHODS: These are well described, and the scales and tools you use seem appropriate, as do the selected variables.

I note there is ethical approval for this study included in another paper – I would recommend adding all aspects of the ethics and consenting process to this paper, so readers don’t need to hunt down another paper to see what was done. This should include how consent for participation was obtained from respondents.

Also, there are two papers cited here that are not in the reference list (and also in a different format). These need to be included in a suitable referencing style in the next version.

RESULTS: These are provided with suitable detail. You itemise the various sub-data sets with commentary, and this is well-organised. Table 2 is a little confusing (‘EBF’ and ‘EFF’ look very similar on reading). Can you add a few words for clarity?

You mention at line 152 the ‘null hypothesis’, which is fine, though I’m not sure you mentioned this specifically as something you would use to interpret the data? For readers without statistical expertise this should perhaps be clarified in the Methods section.

I note that you acknowledge the need to interpret certain findings with caution, and this is helpful.

DISCUSSION/CONCLUSION: The discussion section is valuable and interesting. What I would consider is starting each subsection with a sentence about what you found, and how this relates to current knowledge. Currently, you add your findings to the end. This is only for your consideration, though. I especially like how the findings illustrate some of the tension in diaspora communities, between the culture of their inheritance and the culture of their residence. This is worthy of further study.

I note you say there are four sub-themes, but I couldn’t see the fourth, ‘coping strategies’.  Was this merged with another sub-theme? If so, the first paragraph will need amending. The sub-themes are detailed and do relate broadly to your findings, with some helpful citations. I would advise trimming the text a little – some of the sections are over-written and your (strong) points a little blurred.

I would recommend reworking this sentence: “Although the black culture advocates breastfeeding…” (line 302). ‘Black culture’ may be too generic as a term – and as you point out, attitudes towards breastfeeding can be quite nuanced in the context of HIV.

Finally, I would recommend making stronger recommendations for further research.

REVIEWER RECOMMENDATIONS

1. Introduction:

* Address the two sentences (lines 38-41)

* Define psychosocial distress

* Clarify the sentence lines 41-43

2. Methods and materials:

* Define ACB (line 89)

* Include consenting and ethical components

* Tidy extraneous citations

3. Results:

* Add more commentary for table 2 (EBF/EFF)

* Clarify use of null hypothesis (add to Materials and Methods section)

4. Discussion:

* Find missing sub-theme

* Edit – it’s a little overwritten

* Reword sentence line 302

* Add recommendations for future research

  • Consider adding your finding to the start of each sub-theme (rather than the end)

Author Response

(The authors gave the same response as above.)

Reviewer 4 Report

Well written paper on an area that is often misunderstood and under reported.

Line 121 please identify what IDI means. 

Author Response

(The authors gave the same response as above.)

Reviewer 5 Report

Thank you for your manuscript. I have a few queries and comments that I hope are helpful. My main concern is that I am unclear how the groups are comparable, apart from being of African descent. The social norms and contexts of a woman living in Nigeria including cultural context will be vastly different to a woman living in the USA or Canada. A more details description of the groups would be really helpful to understand this better.

Introduction

line 64 - I am not sure what "craves" refers to.

Your references 15-18 are historical and prior to WHO recommendations on BF. It would be useful to use more recent references to see how this has changed.

Methods

As above, a description of the cohorts will help understand what is common between the groups.

Are the tools used validated in Nigeria and in local language as this is a limitation if not.

It would be helpful if the authors could explain how fear of prosecution has impacted on the truethfulness of the results in the USA/Canada cohorts.

Results

The paragraphs starting on line 147 repeat the tables and the tables, I think are clearer so would summarise these more succinctly. There are fundamental differences between groups with the largest LTFU in the Miami group so understanding the groups is important.

How have women who do not disclose how they feed participate in the discussions? How was their data analysed?

Were the women in USA/Canada who registered as exclusive BF protected from prosecution?

Discussion

Your references mainly refer to the African continent context (and some are historical). It would be useful to cite some HIC literature (if it exists) to show what is simliar across continents.

There is no limitations section to your discussion.

Author Response

(The authors gave the same response as above.)

Round 2

Reviewer 5 Report

My comments have all been addressed.